# Periodic DNA patrolling underlies diverse functions of Pif1 on R-loops and G-rich DNA

Ruobo Zhou[1†], Jichuan Zhang[1,2], Matthew L Bochman[3‡], Virginia A Zakian[3], Taekjip Ha[1,4*]

[1]Center for the Physics of Living Cells, Department of Physics, University of Illinois at Urbana-Champaign, Urbana, United States; [2]Materials Science and Engineering, University of Illinois at Urbana-Champaign, Urbana, United States; [3]Department of Molecular Biology, Princeton University, Princeton, United States; [4]Howard Hughes Medical Institute, University of Illinois at Urbana-Champaign, Urbana, United States

**Abstract** Pif1 family helicases are conserved from bacteria to humans. Here, we report a novel DNA patrolling activity which may underlie Pif1's diverse functions: a Pif1 monomer preferentially anchors itself to a 3′-tailed DNA junction and periodically reel in the 3′ tail with a step size of one nucleotide, extruding a loop. This periodic patrolling activity is used to unfold an intramolecular G-quadruplex (G4) structure on every encounter, and is sufficient to unwind RNA-DNA heteroduplex but not duplex DNA. Instead of leaving after G4 unwinding, allowing it to refold, or going beyond to unwind duplex DNA, Pif1 repeatedly unwinds G4 DNA, keeping it unfolded. Pif1-induced unfolding of G4 occurs in three discrete steps, one strand at a time, and is powerful enough to overcome G4-stabilizing drugs. The periodic patrolling activity may keep Pif1 at its site of in vivo action in displacing telomerase, resolving R-loops, and keeping G4 unfolded during replication, recombination and repair.

*For correspondence: tjha@illinois.edu

**Present address:** [†]Department of Chemistry and Chemical Biology, Harvard University, Cambridge, United States; [‡]Molecular and Cellular Biochemistry Department, Indiana University, Bloomington, United States

**Competing interests:** The authors declare that no competing interests exist.

**Reviewing editor**: Johannes Walter, Harvard Medical School, United States

## Introduction

Approximately, 1% of eukaryotic genes encode DNA or RNA helicases. These enzymes function in nearly all aspects of nucleic acid metabolism in living organisms (*Singleton et al., 2007*; *Lohman et al., 2008*). Although helicases were originally recognized as enzymes that catalyze the strand separation of double-stranded nucleic acids, it is now evident that helicases, as defined by a series of characteristic sequence motifs, may have additional functions aside from unwinding duplexes, including protein displacement from DNA and RNA, the unwinding of G-quadruplex (G4) structures, remodelling of chromatin and ribonucleoprotein complexes, promotion of Holliday junction branch migration and the catalysis of a range of nucleic acid conformational changes (*Singleton et al., 2007*; *Lohman et al., 2008*; *Pyle, 2008*; *Paeschke et al., 2013*).

The Pif1 DNA helicase is an example of a multi-functional helicase. The Pif1 helicase family is a group of 5′→3′ directed, ATP-dependent, super-family (SF) 1B helicases that are evolutionarily conserved from bacteria to humans (*Bochman et al., 2010*; *Paeschke et al., 2013*). The *Saccharomyces cerevisiae* Pif1 helicase (Pif1), the prototypical member of the Pif1 helicase family, plays critical roles in inhibiting telomerase activity at telomeres and double-stranded DNA breaks (DSBs), processing Okazaki fragments, promoting break-induced replication, maintaining mitochondrial DNA, and preventing replication pausing and DSBs at G-quadruplex (G4) motifs (*Boule and Zakian, 2006*; *Bochman et al., 2010*; *Lopes et al., 2011*; *Paeschke et al., 2011, 2013*; *Wilson et al., 2013*). However, the molecular mechanisms responsible for these diverse Pif1 functions remain elusive. For

**eLife digest** Helicases are enzymes that are best known for their ability to separate the two strands of DNA that make up the famous double-helix structure. Many important processes within cells—including the expression of genes as proteins, and the replication of DNA before cell division—rely on DNA molecules being separated in this way. However, these enzymes can perform many other roles that help maintain the integrity of a cell's DNA.

The genetic code is written using four DNA bases—called A, C, G and T—and if a stretch of DNA contains lots of G bases, then one of the strands can loop back upon itself three times to form a structure known as a 'G-quadruplex'. These structures can prevent the expression of genes, and slow the replication of DNA. However, a helicase called Pif1 can unwind G-quadruplexes to allow these activities to continue. This helicase is found in many organisms, from bacteria to humans, and carries out multiple functions for a cell. However, the exact mechanisms underlying these activities are unknown.

Now, Zhou et al. have used biophysical techniques to reveal that individual Pif1 proteins bind to single-stranded overhangs at one end of a DNA molecule. Pif1 also binds to forks in DNA where the double helix separates into two single strands. And once Pif1 has bound to the DNA, it works to 'reel in' the overhang or a single strand, one base at a time. This activity can unwind a G-quadruplex, and individual Pif1 proteins will patrol DNA to keep this structures unwound without unraveling the double helix itself. Separating the two strands of DNA actually needs multiple Pif1 proteins to join and work together.

As it patrols, Pif1 also displaces other proteins from DNA and removes unusual, and potentially harmful, structures in DNA (such as RNA molecules that have displaced one of the strands of DNA double helix). The next challenge will be to address important questions that remain unanswered including: how does Pif1 recognize DNA structures and change its activity; and how does it coordinate with other proteins that target the same structures?

example, Pif1 is known to unwind RNA/DNA hybrids better than dsDNA but its mechanistic basis is unknown (*Boule and Zakian, 2007*). In addition, how Pif1 may selectively function on certain DNA structures such as stalled replication forks or G4 structures is unclear.

Here, we show that a Pif1 monomer is preferentially recruited to 3′ ss-dsDNA junctions and induces repetitive DNA looping that is tightly coupled to its translocation activity powered by its ATPase. This periodic DNA patrolling activity of a Pif1 monomer can be used to unwind RNA–DNA hybrids in its path but not DNA–DNA duplexes because DNA unwinding requires the cooperation of multiple Pif1 monomers. Furthermore, we show that a Pif1 monomer has the capacity to unwind an intramolecular G4 structure in its patrolling path with near unity yield. This novel activity would keep the enzyme at its site of in vivo action in displacing telomerase from 3′ ssDNA ends (*Boule et al., 2005*), resolving biologically relevant 'R-loops' (*Aguilera and Garcia-Muse, 2012*), and keeping G4 DNA sequences unfolded during DNA replication, recombination and repair (*Paeschke et al., 2011*; *Wilson et al., 2013*).

## Results

### A Pif1 monomer scans 3′ ssDNA overhang periodically

We employed single-molecule (sm)FRET assays with total internal reflection (TIR) fluorescence microscopy (*Roy et al., 2008*) to study the ATP-dependent 5′ to 3′ ssDNA translocation activity of *Saccharomyces cerevisiae* Pif1 (*Galletto and Tomko, 2013*). We first measured the binding constant ($K_D$) of Pif1 to a partial duplex (pd) DNA with a 3′ 40-nt Poly(dT) overhang (*Figure 1A*). The donor (Cy3) and the acceptor (Cy5) were attached to the overhang separated by 16 nt so that a relatively high FRET efficiency ($E_{FRET}$) of ~0.58 was observed (*Figure 1B*) due to the high flexibility of ssDNA (*Murphy et al., 2004*). Upon addition of Pif1, a new population with $E_{FRET} = 0.37$ appeared because protein binding stretches ssDNA (*Yang et al., 2013*). Half of the DNA was occupied by Pif1 at 7 ± 2 nM Pif1 concentration after 2-min incubation, and Pif1 dissociation was minimal 30 min after washing out unbound proteins (*Figure 1—figure supplement 1*). Unless specified otherwise, all experiments were

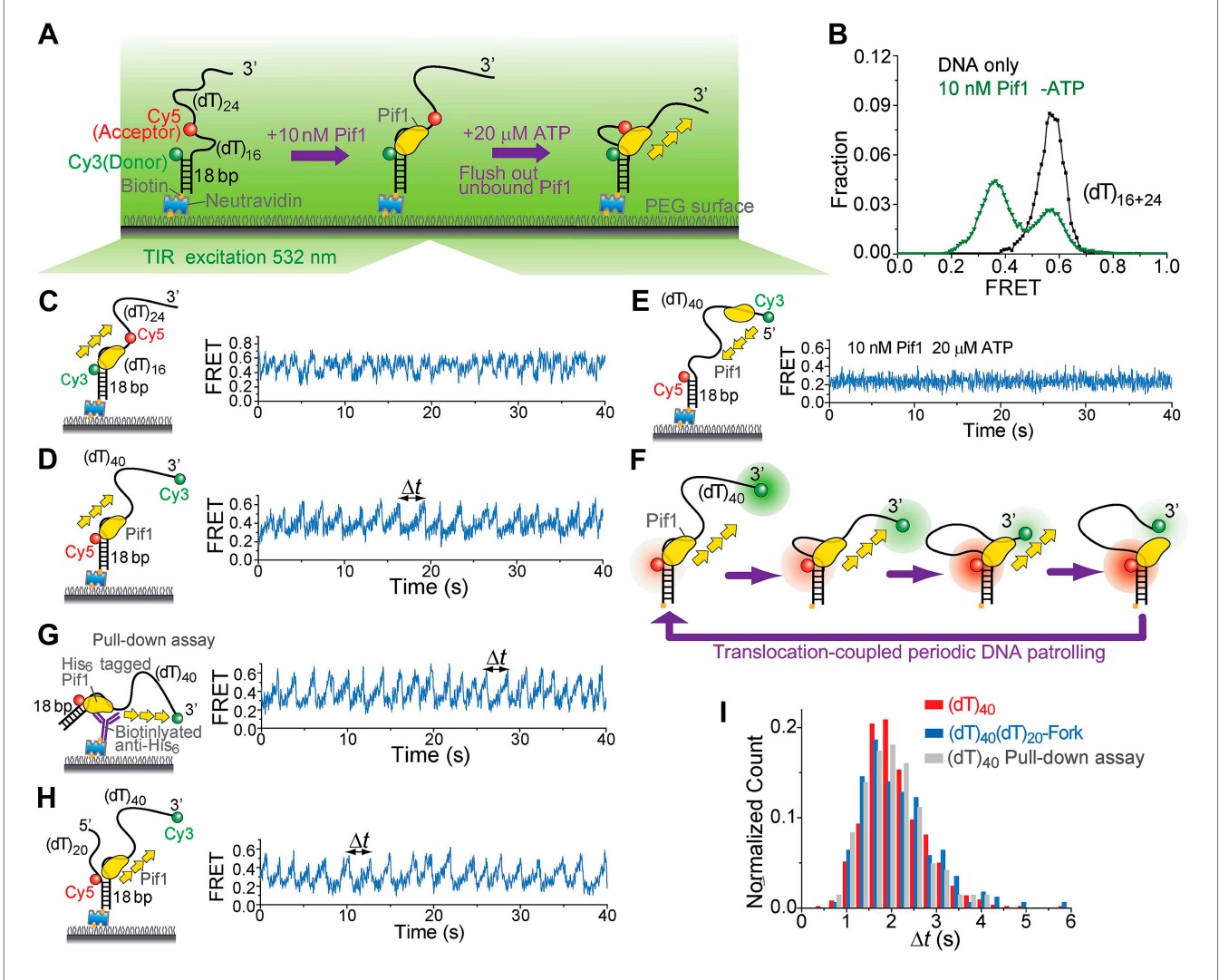

**Figure 1**. Periodic DNA patrolling by a Pif1 monomer. (**A**) A schematic representation of reaction steps for smFRET measurements under the monomer condition. (**B**) smFRET histograms obtained before and after adding 10 nM Pif1 in the absence of ATP. Peaks centered at $E_{FRET}$ = 0.58 and 0.37 were observed represent 0 and 1 Pif1 molecules binding to DNA, respectively. (**C** and **D**), smFRET-time traces showing repetitive translocation of a Pif1 monomer on $(dT)_{40}$. The labeling positions of Cy3 and Cy5 differ in (**C**) vs (**D**). (**E**) smFRET-time trace for a DNA substrate containing a 5' Poly(dT) ssDNA overhang (referred to as $(dT)_{40}$-5') showing no sawtooth pattern. 10 nM Pif1 and 20 µM ATP were added. (**F**) A schematic representation of the repetitive looping model. A Pif1 monomer remains bound at the 3' ss-dsDNA junction and reels in the 3' ssDNA overhang. (**G**) A smFRET-time trace showing repetitive translocation of a surface-immobilized Pif1 monomer on non-biotinylated $(dT)_{40}$. (**H**) smFRET-time trace showing repetitive looping by a Pif1 monomer on a forked DNA substrate. (**I**) Histograms of the time interval of repetition ($\Delta t$).

The following figure supplements are available for figure 1:

**Figure supplement 1**. Pif1 binding to $(dT)_{16+24}$ DNA.

**Figure supplement 2**. Pif1 binding to $(dT)_{16+24}$, $(dT)_{40}$, and $(dT)_{21}$.

**Figure supplement 3**. Investigation of Pif1's periodic patrolling activity using protein-induced fluorescence enhancement (PIFE).

performed in the absence of free proteins after 2-min pre-incubation with 10 nM Pif1 followed by wash (referred to as 'monomer condition').

When ATP was added, the smFRET histogram became broader (*Figure 1—figure supplement 1A*), and smFRET-time traces exhibited periodic fluctuations between $E_{FRET}$ = 0.37 and 0.58 (*Figure 1C*),

likely due to Pif1 translocation on and off the 16-nt segment separating the fluorophores. When we repositioned the fluorophores near the two ends of the $(dT)_{40}$ region (the construct is referred to as $(dT)_{40}$), smFRET-time traces showed a periodic sawtooth-shaped pattern where FRET gradually increases until it abruptly drops to a low level and repeats the process many times (*Figure 1D*). Observation of the sawtooth pattern required both Pif1 and ATP (*Figure 1—figure supplement 2*), and the sawtooth pattern was not observed with a 5′ overhang (*Figure 1E*). We hypothesized that Pif1 remains anchored to the 3′ ss-dsDNA junction while translocating on and reeling in the 3′ overhang, extruding a ssDNA loop (*Figure 1F*). To test this hypothesis, we used the fact that Cy3 gets brighter when a protein is in close proximity (*Fischer et al., 2004*; *Hwang et al., 2011*). A sawtooth pattern was observed when Cy3 was placed at the 3′ ssDNA end but not at the ss-dsDNA junction, supporting the model that Pif1 approaches the 3′ end while remaining anchored to the junction (*Figure 1—figure supplement 3*). To ascertain the monomeric state of Pif1, Pif1 was pulled down to the surface (*Jain et al., 2011*) using an antibody against the Histidine₆-tag (*Figure 1G*). Pif1 pulled-down in this way should maintain its monomeric state in solution (*Lahaye et al., 1993*; *Barranco-Medina and Galletto, 2010*). The same sawtooth patterns were observed when 5 nM $(dT)_{40}$ was added with ATP (*Figure 1G*), indicating that a Pif1 monomer is responsible for the observed repetitive DNA looping that we term here as 'periodic patrolling'. The presence of a 5′ overhang in addition to the 3′ overhang did not inhibit periodic patrolling (*Figure 1H*). The histograms of the patrolling period, $\Delta t$, were nearly identical among all three schemes (>50 molecules each; *Figure 1I*), yielding an average patrolling period $\tau = 2.1 \pm 0.1$ s.

## DNA translocation properties

Pif1 monomers are ATP-dependent translocases on ssDNA with 5′ to 3′ directionality (*Galletto and Tomko, 2013*). Periodic patrolling slowed down as we lowered [ATP] from 500 to 5 µM (*Figure 2A*). Plotting the inverse of $\tau$ vs [ATP] and fitting to the Michaelis–Menten equation, we obtained $K_M = 110 \pm 17$ µM (*Figure 2B*). We examined Pif1's periodic patrolling on 3′ Poly(dT) overhangs of varying lengths ($N$ = 32, 40, 56, and 72 nt; referred to as $(dT)_N$ at a fixed [ATP] (20 µM) and found that $\tau$ increased linearly with the overhang length (*Figure 2C,D*), showing that the patrolling is indeed coupled to ssDNA translocation with a speed of 13 ± 2 nt/s at 20 µM ATP, corresponding to 85 ± 13 nt/s at saturated [ATP], consistent with previous ensemble studies (*Galletto and Tomko, 2013*; *Ramanagoudr-Bhojappa et al., 2013*).

To determine the kinetic step size for Pif1 translocation, we assumed that Pif1 takes $n$ hidden irreversible Poissonian steps with identical rate $k$ in each cycle of duration $\Delta t$. A $\Delta t$ histogram obtained from a single Pif1 that showed hundreds of cycles on $(dT)_{32}$ was fit with the $\Gamma$-distribution, $(\Delta t)^{n-1}\exp(-k \cdot \Delta t)$ (*Figure 2E*) (*Park et al., 2010*). The fit yielded $n = 26 \pm 3$, $k = 25 \pm 3$ s$^{-1}$ for one molecule of $(dT)_{32}$, suggesting 26 or more steps required to translocate over 32 nt, which supports a 1-nt step size. There is variation of Pif1 translocation speed among different Pif1 molecules, but a 1-nt step size could be consistently determined (*Figure 2—figure supplement 1*). Related to our findings, a recent ensemble study proposed a 1-bp step size of dsDNA unwinding by Pif1 (*Ramanagoudr-Bhojappa et al., 2013*).

## A Pif1 monomer unwinds RNA–DNA hybrids but not dsDNA

It has been proposed that Pif1 inhibits telomerase activity by unwinding the RNA–DNA hybrid formed between telomerase RNA and the 3′ end of telomeric DNA (*Boule et al., 2005*; *Boule and Zakian, 2007*) and that Pif1 removes R-loops, that is, RNA–DNA hybrids that occur naturally during replication and transcription (*Boule and Zakian, 2007*; *Aguilera and Garcia-Muse, 2012*). Whether a monomer or dimer of SF1/SF2 helicases is responsible for nucleic acid unwinding has been a subject of debate (*Singleton et al., 2007*; *Lohman et al., 2008*), so we asked if a patrolling Pif1 monomer can unwind dsDNA or RNA–DNA hybrids. We designed gapped substrates by placing an additional 31-bp DNA–DNA or RNA–DNA duplex next to the 3′ end of the $(dT)_{40}$ substrate (referred to as DD31 and RD31, respectively; *Figure 3A*). If Pif1 fully unwinds the duplex during its periodic patrolling, the Cy3-labeled strand should be released from the surface, resulting in a decrease of the number of Cy3 spots over time. DD31 unwinding was not detectable under the monomer condition but was observed at 100 nM Pif1, likely due to multiple Pif1 proteins acting in concert (*Figure 3A*). In contrast, efficient unwinding of RD31 was detected under both conditions (*Figure 3A*). Under the monomer condition, unwinding curves for RD31 can be fit to a single exponential function, yielding the unwinding times $t_{unwind} = 6.3 \pm 0.9$ and $1.6 \pm 0.2$ min at 20 µM and 1 mM ATP, respectively (*Figure 3—figure supplement 1C*). smFRET-time

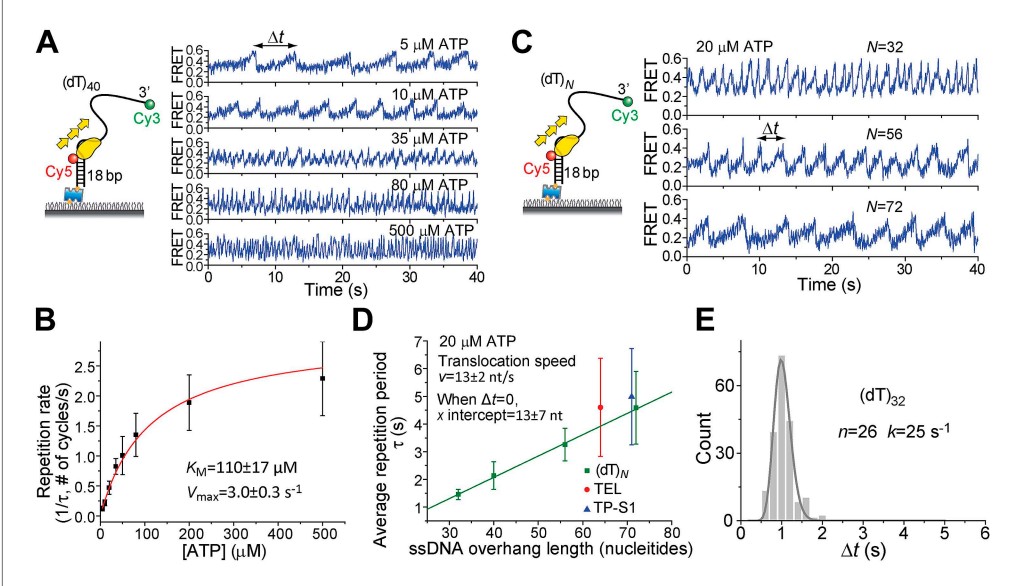

**Figure 2**. ATP and ssDNA length dependence of the repetition period and step size determination. (**A**) smFRET time traces showing Pif1 repetitive looping on $(dT)_{40}$ at varying ATP concentrations. (**B**) Michaelis–Menten fit of repetition rate $(1/\tau)$ vs ATP concentration. Error bars denote SD. Errors in the fit results are shown in SEM. (**C**) smFRET time traces of repetitive looping on $(dT)_N$ at 20 µM ATP. To obtain the average translocation speed, $v$, we examined RL with pdDNA containing 3′ Poly(dT) overhangs of varying lengths ($N$ = 32, 40, 56, and 72 nt; referred to as $(dT)_N$). (**D**) Average repetition period, $\tau$, vs 3′ ssDNA overhang length. The error bar denotes SD. Errors in the fit result are SEM. (**E**) $\Delta t$ histogram obtained from a single Pif1 monomer showing 190 repetitive looping events on $(dT)_{32}$. The solid line is a fit to the Γ-distribution.

The following figure supplements are available for figure 2:

**Figure supplement 1**. Histograms of the repetition period, $\Delta t$, for Pif1 periodic patrolling on $(dT)_N$.

---

traces taken under the monomer condition showed that Pif1 performs periodic patrolling on both DD31 and RD31 (**Figure 3B**, **Figure 3—figure supplement 1A**) with an average period of $\tau$ = 2.4 ± 0.1 s at 20 µM ATP (**Figure 3C**, **Figure 3—figure supplement 1A**). Our data hence suggest that a Pif1 monomer takes about 200 cycles on average (= 6.3 min/2.4 s) to unwind the 31 bp RNA–DNA hybrid, whereas dsDNA unwinding requires the cooperation of multiple Pif1 monomers or protein oligomerization on DNA. At 100 nM Pif1, FRET fluctuations became irregular (**Figure 3B**), likely due to the binding of multiple Pif1 monomers to the gapped substrate at this Pif1 concentration. The dsDNA unwinding speed previously reported was much higher (~0.3 s for unwinding 16 bp dsDNA) (**Ramanagoudr-Bhojappa et al., 2013**), either because they pre-assembled a functional helicase unit on the DNA before adding ATP to start the reaction or in that study, or due to a limited processivity of dsDNA unwinding by a Pif1 oligomer.

## 3′-single-stranded/duplex DNA junctions are preferential loading sites for Pif1 monomers

In a gapped DNA substrate, both 3′- and 5′-ss-dsDNA junctions are present for Pif1 loading. To determine Pif1's preferential binding site on a gapped substrate, we used two labeling schemes: (1) $(dT)_{32+18}$-S1: FRET probes were attached to the middle and 5′ end of the gap region; (2) $(dT)_{32+18}$-S2: the probes were attached to the middle and 3′ end of the gap region so that FRET reports on Pif1 binding to the respective ssDNA segments. At 10 nM Pif1, a FRET decrease was observed only for $(dT)_{32+18}$-S1, whereas at 100 nM Pif1, FRET decreased for both schemes (**Figure 4**), indicating that Pif1 prefers to load at 3′-ss-dsDNA junctions and in agreement with our observation that at 100 nM Pif1, binding of multiple Pif1 monomers to a gapped substrate abolishes the regular sawtooth pattern (**Figure 3B**, **Figure 3—figure supplement 1C**).

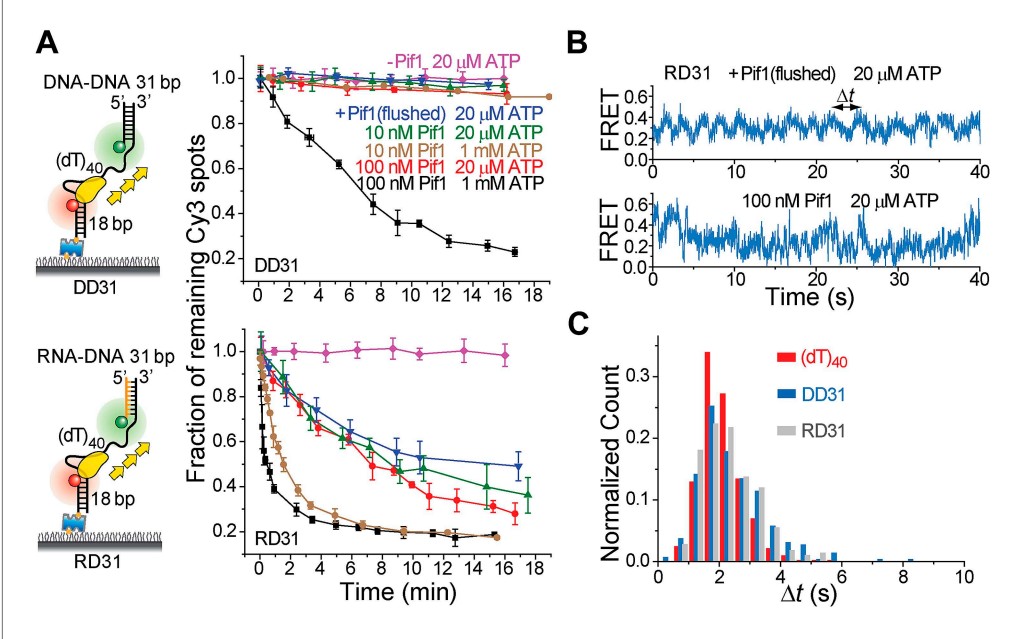

**Figure 3**. Periodic DNA patrolling of a Pif1 monomer can unwind RNA-DNA hybrids but not dsDNA. (**A**) Unwinding curves for DD31 (top) and RD31 (bottom) obtained under different conditions. (**B**) A smFRET time trace of RD31 obtained under the monomer condition shows Pif1 repetitive looping (top) and a smFRET time trace of RD31 obtained at 100 nM Pif1 shows irregular FRET fluctuations (bottom). (**C**) $\Delta t$ histograms for Pif1 periodic patrolling on $(dT)_{40}$, DD31, and RD31.

The following figure supplements are available for figure 3:

**Figure supplement 1**. Translocation and unwinding activity of Pif1 on gapped substrates.

## Periodic patrolling by a Pif1 monomer unwinds G4 DNA structures and keep them unfolded

Pif1 is also a potent unwinder of G-quadruplexes (G4s) (*Paeschke et al., 2011*), which are held together by noncanonical Hoogsteen G–G base pairs and stabilized by monovalent cations (*Huppert, 2008*). Intramolecular G4 motifs (four runs of 'GGG', with loops of 1–25 nt between them of any sequence) are prevalent in eukaryotic genomes (gene promoter regions, telomeres, etc) (*Bochman et al., 2012*), and G4 formation may slow replication and regulate transcription (*Paeschke et al., 2011*; *Bochman et al., 2012*). Growing evidence indicates that G4 structures indeed form in vivo (*Lipps and Rhodes, 2009*; *Paeschke et al., 2011, 2013*; *Biffi et al., 2013*).

To test whether a patrolling Pif1 monomer can unwind intramolecular G4 DNA, we placed a 31-nt standard G4 sequence (termed TP) from the mouse immunoglobulin locus (*Paeschke et al., 2013*) at the 3' end of the $(dT)_{40}$ region (referred to as TP-S1; *Figure 5A*). Cy3 and Cy5 were placed at the two ends of TP so that a folded G4 structure would yield high FRET. As a control, we replaced the 31-nt TP sequence with Poly(dT) of the same length (*Figure 5B*). We confirmed G4 formation by performing K+ titration (*Figure 5—figure supplement 1*). In 5 mM $Mg^{2+}$ and 60 mM K+, TP-S1 gave a single population of $E_{FRET}$ = ~0.75 for the folded G4 structure in contrast to the control that gave $E_{FRET}$ = ~0.3 (*Figure 5A,B*). Addition of 10 nM Pif1 alone did not alter the smFRET histogram of TP-S1 (*Figure 5A*), suggesting Pif1 cannot unwind G4 in the absence of ATP, when K+ is present.

Upon addition of ATP, a lower FRET species appeared, indicating G4 unwinding. Remarkably, smFRET-time traces showed a periodic G4 unwinding pattern. In each unwinding cycle ($\Delta t$), $E_{FRET}$ starts at ~0.75 and slowly decreases to ~0.3, followed by an abrupt increase back to 0.75 (*Figure 5C*). For a modified labeling scheme of placing Cy3 at 3' end of TP sequence and Cy5 at 3' ss-dsDNA junction (TP-S2; *Figure 5—figure supplement 2B*), smFRET-time traces resembled the sawtooth patterns seen for $(dT)_N$, indicating that before unwinding G4, a Pif1 monomer patrols the $(dT)_{40}$ region. The

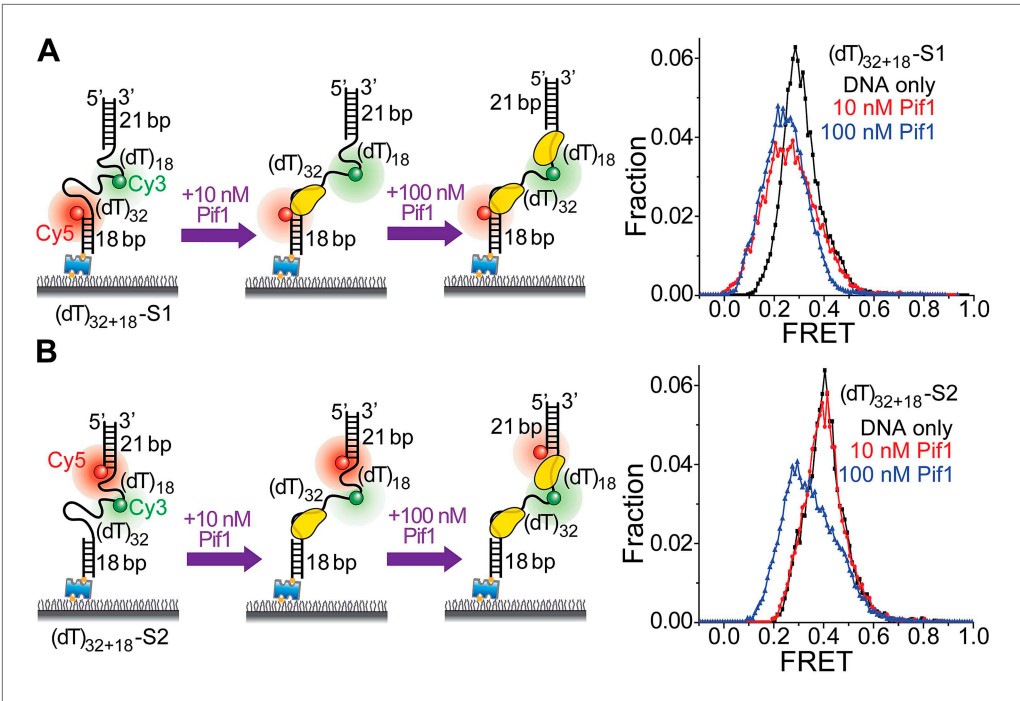

**Figure 4.** Pif1 monomers are preferentially recruited to 3′ ss-dsDNA junctions. (**A** and **B**) Schematic representations of reaction steps to identify the preferential Pif1 binding site to a gapped DNA substrate. The two gapped DNA substrates used ((dT)$_{32+18}$-S1 and (dT)$_{32+18}$-S2) only differ in Cy5 labeling position. smFRET histograms were determined at different Pif1 concentrations.

observation indicates that Pif1 unwinds G4 with near unity yield on each encounter. The average period of the unwinding pattern, τ, depends on [ATP] with a $K_M$ of 118 ± 7 μM (***Figure 5C,D***, ***Figure 5—figure supplement 2A***) and is close to the average patrolling period of Pif1 on a 3′ Poly(dT) overhang of the same overhang length (71 nt; ***Figure 2D***). The same periodic unwinding pattern was observed with pulled down monomeric Pif1 molecules (***Figure 5G***). The (dT)$_{40}$ region is necessary for G4 unwinding in K$^+$ solution (***Figure 5F***) but not in Na$^+$ solution (***Figure 5—figure supplement 3***), indicating that the G4 structure is easier to be unwound in Na$^+$ than in K$^+$. In Na$^+$ solution, addition of 10 nM Pif1 led to the unwinding of the G4 structure without an adjacent ssDNA region, even in the absence of ATP (***Figure 5—figure supplement 3A***), suggesting that Pif1 binding alone may unfold TP G4 in Na$^+$. This is consistent with the known order of cations' ability to stabilize G4: K$^+$>Na$^+$>Li$^+$ (***Simonsson, 2001***), further confirming that the secondary structure formed by TP sequence is indeed G4.

We next tested another well-studied G4 sequence from human telomeres (***Ying et al., 2003***; ***Lee et al., 2005***) by replacing TP with A(GGGTTA)$_3$GGGTT (referred to as TEL). K$^+$ and Na$^+$ titrations with and without Mg$^{2+}$ indicated that Mg$^{2+}$ facilitates TEL G4 formation both in K$^+$ and Na$^+$ (***Figure 5—figure supplement 4***). Similar smFRET histograms (***Figure 5H***) and periodic unwinding time traces (***Figure 5I***) were obtained, implying that the periodic G4 unwinding by a Pif1 monomer may be general and not specific to sub-species of G4 structures. Similar to the observations for TP, deletion of the ssDNA region at the 5′ end of TEL (***Figure 5J***) inhibited Pif1 loading and G4 unwinding in K$^+$. The inhibitory effect remained in Na$^+$ but was largely reduced in Li$^+$ (***Figure 5—figure supplement 3C***), likely due to the inability of Li$^+$ to stabilize TEL G4.

Addition of a 21-bp DNA duplex next to the 3′ end of TEL (***Figure 5—figure supplement 5***) did not affect the periodic G4 unwinding, indicating that even though a Pif1 monomer repetitively unwinds G4 DNA with a very high yield, it does not go beyond it to unwind dsDNA. This intrinsic activity of a Pif1 monomer would be useful in keeping G4 DNA unfolded until other DNA metabolic enzymes can take over without causing the side effect of uncontrolled dsDNA unwinding.

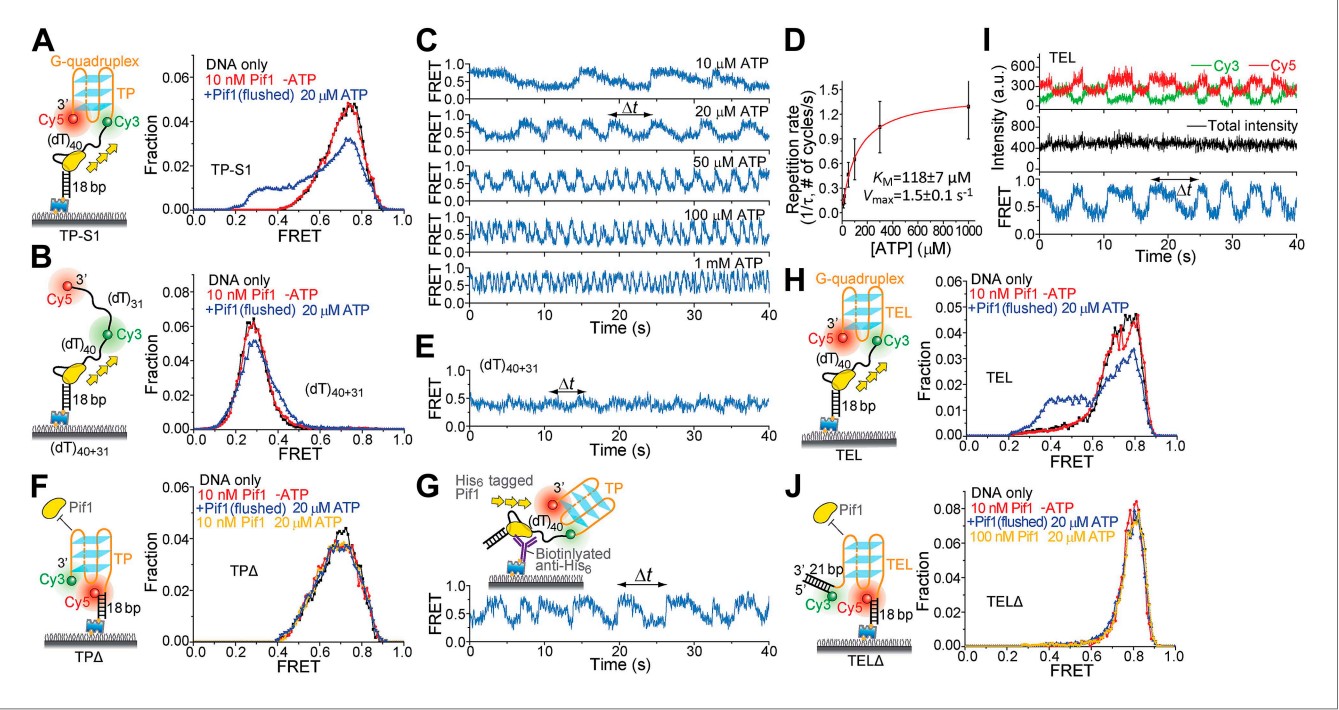

**Figure 5**. Periodic DNA patrolling of a Pif1 monomer can unwind G4 structures. (**A** and **B**) smFRET histograms of TP-S1 (**A**) and $(dT)_{40+31}$ (**B**) obtained under different conditions. (**C**) A smFRET time trace showing Pif1 repetitive unwinding on TP-S1 at varying ATP concentrations. (**D**) Michaelis–Menten fit of the repetition rate $(1/\tau)$ vs ATP concentration. Error bars denote SD. Errors in the fit results are SEM. (**E**) A smFRET time trace showing Pif1 periodic patrolling on $(dT)_{40+31}$. $E_{FRET}$ changes periodically between ~0.3 and 0.4, likely due to Pif1 translocation on and off the $(dT)_{31}$ segment separating Cy3 and Cy5. (**F**) smFRET histograms for a DNA substrate devoid of the $(dT)_{40}$ region (referred to as TPΔ) obtained under different conditions. The G4 structure is not unwound when the ssDNA region is absent, likely due to lack of Pif1 loading. (**G**) A smFRET time trace showing repetitive G4 unwinding of a surface-immobilized Pif1 monomer on non-biotinylated TP-S1. (**H** and **I**) smFRET histograms (**H**) and time traces (**I**) of TEL. (**J**) smFRET histograms of TELΔ obtained under different conditions.

The following figure supplements are available for figure 5:

**Figure supplement 1**. K⁺ titrations to study G4 formation for the TP sequence in the presence and absence of Mg²⁺.

**Figure supplement 2**. Monomeric Pif1 repetitively unwinds TP G4 in the presence and absence of BRACO19.

**Figure supplement 3**. Pif1 binding to TPΔ and TELΔ in the presence of Na⁺ or Li⁺.

**Figure supplement 4**. K⁺ and Na⁺ titrations to study G4 formation for the TEL sequence in the presence and absence of Mg²⁺.

**Figure supplement 5**. Presence of a DNA duplex next to the 3′ end of the G4 sequence does not affect Pif1's periodic G4 unwinding activity.

**Figure supplement 6**. Dwell time histograms of each unwinding step and histogram of G4 refolding time built from TP-S1.

**Figure supplement 7**. Δt histograms of different DNA substrates.

## G4 unwinding activity is specific to Pif1

To test if the periodic patrolling activity alone is sufficient to unwind G4, we analyzed the *Bacillus stearothermophilus* PcrA, which also exhibits periodic DNA patrolling but with reverse translocation directionality (*Park et al., 2010*). We designed new versions of $(dT)_{40}$ and TEL with reverse polarity (*Figure 6A,B*). Under monomer conditions (1 nM PcrA) (*Park et al., 2010*), PcrA showed robust periodic patrolling, but no G4 unwinding was detected (*Figure 6C,D*). Even at 100 nM, PcrA efficiently unwound dsDNA (*Figure 6E*), but efficient G4 unwinding was not detected. These data suggest that

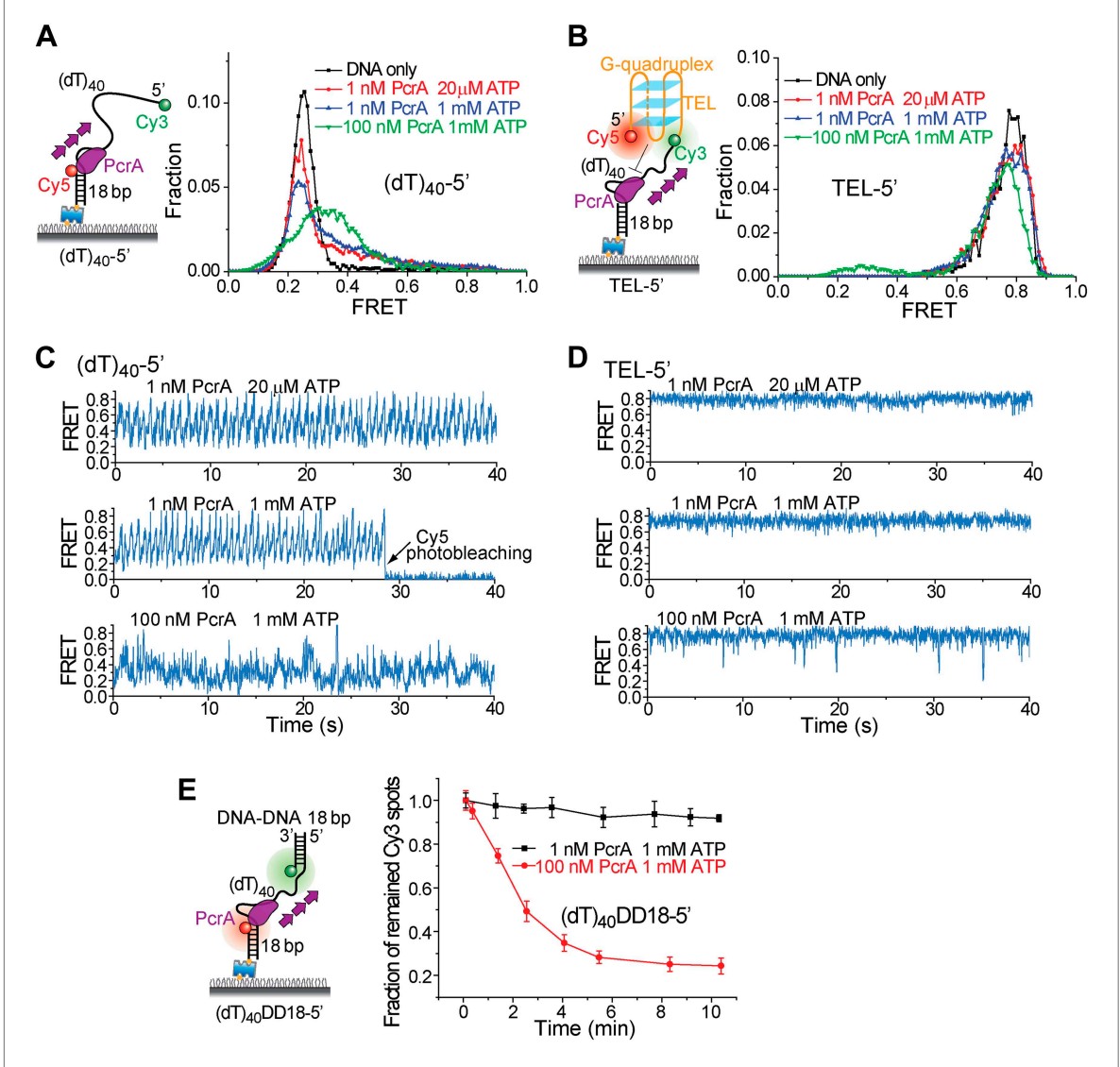

**Figure 6**. Periodic DNA patrolling of a PcrA monomer cannot unwind G4 structures. (**A**) smFRET histograms of $(dT)_{40}$-5' obtained for DNA only or at 1 or 100 nM PcrA (ATP was added). (**B**) smFRET histograms of TEL-5' obtained for DNA only or at 1 or 100 nM PcrA (ATP was added). (**C**) Single molecule time traces showing PcrA dynamics on $(dT)_{40}$-5' obtained at 1 or 100 nM PcrA. When 1 nM PcrA was added, periodic sawtooth patterns were observed for periodic DNA patrolling by a PcrA monomer as previously demonstrated (**Park et al., 2010**). However, when 100 nM PcrA was added, the sawtooth patterns were disrupted, and only irregular FRET dynamics were observed, indicating the binding of multiple PcrA monomers per DNA or protein oligomerization. (**D**) Single molecule time traces of TEL-5' obtained under the same conditions as in (**C**). No efficient G4 unwinding was observed. (**E**) The dsDNA unwinding activity at 1 and 100 nM PcrA. At 1 nM PcrA, no efficient unwinding was observed in 10 min at 1 nM PcrA, whereas efficient unwinding was observed at 100 nM PcrA.

having the periodic patrolling activity per se is not sufficient for G4 unwinding, and robust G4 unwinding is a specific property of Pif1.

## A Pif1 monomer unwinds G4 DNA in three discrete steps

Upon close examination of the G4 unwinding phase, we noticed that the FRET decrease occurs in three discrete steps (**Figure 7A**). We determined the average $E_{FRET}$ values and dwell times of each step ($\Delta t_1$, $\Delta t_2$, $\Delta t_3$ and $\Delta t_4$) using an unbiased step-finding algorithm (**Kerssemakers et al., 2006**) (**Figure 7B**). The distributions of average $E_{FRET}$ values of the steps thus identified (100–200 molecules each) revealed four peaks, showing that G4 unwinding indeed occurs in three steps (**Figure 7C**,

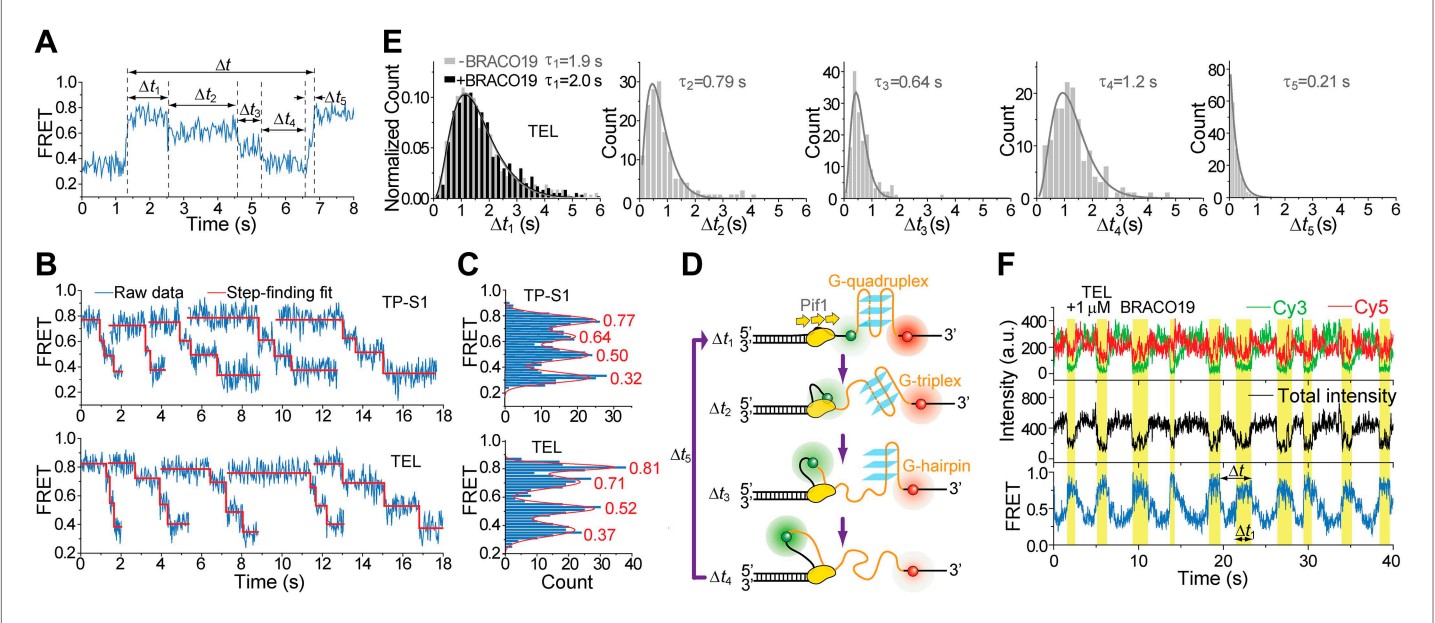

**Figure 7**. Pif1 unwinds G4 in three discrete steps. (**A**) A representative unwinding cycle showing a stepwise pattern for G4 unwinding followed by fast G4 refolding. The dwell times of different parts of the cycle, $\Delta t_1$, $\Delta t_2$, $\Delta t_3$, $\Delta t_4$ and $\Delta t_5$, are indicated. (**B**) The average $E_{FRET}$ values and dwell times at each step were determined by an automated step-finding algorithm (red). (**C**) Distributions of the average $E_{FRET}$ values determined from many unwinding cycles collected from 100 to 200 molecules. (**D**) A proposed model for G4 unwinding involving two intermediates. (**E**) Dwell time histograms of each unwinding step and histogram of G4 refolding time $\Delta t_5$ (gray). A histogram of $\Delta t_1$ (**F**) obtained in the presence of BRACO19 is shown in black. Solid lines are Γ-distribution fits. (**F**) Single molecule time traces showing repetitive G4 winding by a Pif1 monomer in the presence of BRACO19. The total fluorescence intensity of Cy3 and Cy5 fluctuates periodically between two levels, and the low intensity level (yellow regions, $\Delta t_1$) coincides well with the folded G4 state.

*Figure 5—figure supplement 5D*). The highest FRET state was assigned to the folded G4 DNA and the lowest FRET state to the fully unfolded state. The two additional FRET states likely represent partially folded states (G-hairpin and G-triplex), as previously proposed based on ensemble kinetic and simulation studies (*Mashimo et al., 2010*; *Zhang and Balasubramanian, 2012*) (*Figure 7D*). The dwell time histogram of each FRET state exhibits a Γ-distribution (*Figure 7E*, *Figure 5—figure supplement 6*), suggesting that Pif1 reels in DNA in multiple 1-nt steps before the next strand is unraveled and so on. The refolding occurred rapidly with a single rate-limiting step (~0.2 s refolding time, see the $\Delta t_5$ histogram) even for the TP sequence with 7 nt of intervening ssDNA between G repeats (*Mashimo et al., 2010*; *Zhang and Balasubramanian, 2012*), and the subsequent three-step unfolding events by Pif1 indicate that the G4 structure is indeed reestablished rapidly. Instead of resolving a G4 structure once and leaving the DNA, which would allow the intramolecular G4 structure to form again, Pif1 stays in proximity and continually resolves the G4 DNA using its periodic patrolling activity.

Finally, we found that Pif1's patrolling activity is strong enough to unwind G4 structures stabilized by a small compound, BRACO19 (*De Cian et al., 2007*). smFRET-time traces of TEL (*Figure 7F*) and of TP-S1 (*Figure 5—figure supplement 2C*, *Figure 5—figure supplement 7*) showed periodic unwinding patterns in the presence of 1 µM BRACO19 with a nearly identical τ value (~4.7 s; *Figure 5—figure supplement 7*) to what was obtained without BRACO19. Periodic fluctuations of the total fluorescence intensity caused by fluorescence quenching by BRACO19 confirmed the binding of the compound to G4 (*Figure 5—figure supplement 2D*, *Figure 7E*).

## Discussion

Our study establishes a novel DNA patrolling activity by a Pif1 monomer. We discovered that monomeric Pif1 is preferentially recruited to 3'-ss-dsDNA junctions and reels in ssDNA while staying at the junction, repeatedly looping out a ssDNA segment. This periodic behavior is ATP dependent and tightly coupled to Pif1's translocase activity. Previous single-molecule studies have shown that *B. stearothermophilus* PcrA and *Escherichia coli* UvrD, two bacterial SF 1A helicases, are both

preferentially recruited to 5′-ss-dsDNA junctions and exhibit similar activities with reverse translocation directionality (i.e., 3′→5′) (**Park et al., 2010**; **Tomko et al., 2010**). In addition, we showed that Pif1's patrolling activity persists in forked and gapped DNA substrates containing a 3′-ss-dsDNA junction, raising the possibility that it may happen when Pif1 binds to various types of DNA intermediates during DNA replication, repair, and recombination where a 3′-ss-dsDNA junction is present.

**Figure 8** illustrates several possible biological contexts in which Pif1's periodic patrolling activity may play important roles (**Figure 8**). First, the patrolling activity of a Pif1 monomer can be used to unwind 'R-loops' (RNA-DNA hybrids) (**Figure 8A**). Growing evidence indicates that R-loops occur much more frequently during replication and transcription than previously foreseen, and persistent R-loop formation in cells is a potential source of genome instability including mutations, recombination, chromosomal rearrangements, and chromosome loss (**Aguilera and Garcia-Muse, 2012**). We showed that a Pif1 monomer can utilize its patrolling activity to remove the RNA strand from 31-bp RNA–DNA hybrids through ~200 cycles (and fewer cycles may be needed for shorter heteroduplexes). In contrast, dsDNA unwinding required the cooperation of multiple Pif1 monomers. Therefore, it is possible that Pif1's periodic patrolling activity contributes to the cell's ability to remove deleterious R-loops. Second, periodic patrolling may be responsible for Pif1's telomerase removal activity from 3′ ssDNA ends found at telomeres or DSBs (**Boule et al., 2005**) (**Figure 8B**). Many previous studies show that the translocation activity of monomeric helicases can help displace proteins from DNA (**Lohman et al., 2008**). In fact, PcrA utilizes periodic patrolling activity to dismantle potentially deleterious RecA filaments from 5′ ssDNA ends (**Park et al., 2010**). As it has been proposed that Pif1 inhibits telomerase by unwinding the RNA–DNA hybrid formed between the telomerase RNA and the telomeric DNA end (**Boule and Zakian, 2007**), Pif1's periodic patrolling activity, which is capable of unwinding RNA–DNA heteroduplexes, may be responsible for the Pif1-mediated telomerase inhibition at 3′ ssDNA ends. Third, periodic patrolling can be used to keep G4 DNA unfolded at transcription bubble (**Figure 8A**), at DSBs or telomeres (**Figure 8C**) or on the lagging and/or leading strand during replication (**Figure 8D,E**). We showed that periodic patrolling by a Pif1 monomer can unwind intramolecular G4 DNA structures at every patrolling cycle. Because, as shown here, an unwound G4 DNA refolds immediately (~0.2 s), the periodic patrolling activity of Pif1 due to its anchoring at the junction is critical to ensure that G4 structures are kept resolved. In gapped DNA substrates, the patrolling activity can keep G4 sequences unfolded in the ssDNA gap while leaving the

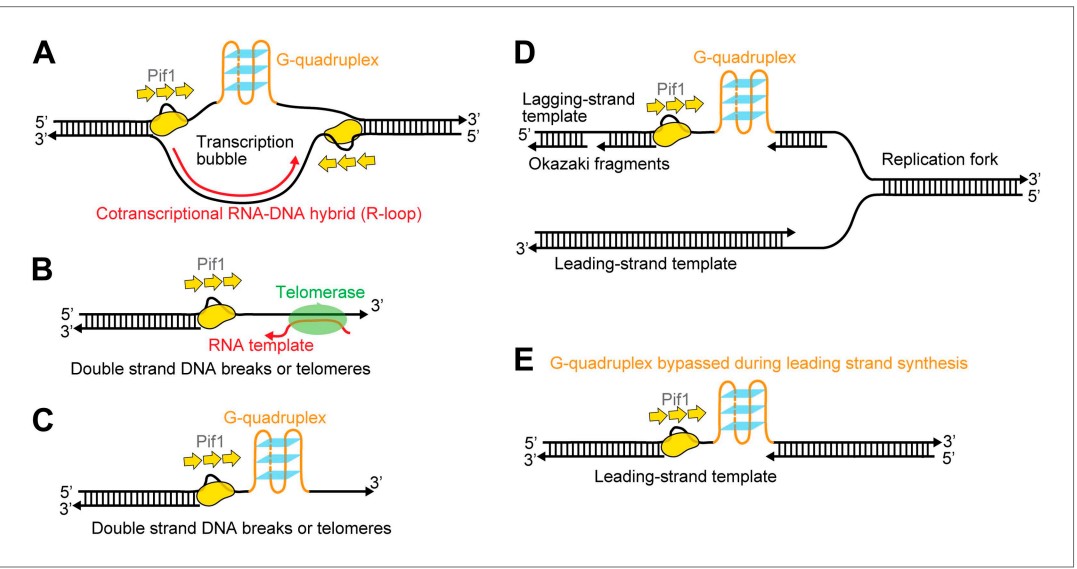

**Figure 8**. Possible cellular sites for the patrolling activity of a Pif1 monomer. The periodic patrolling activity may keep Pif1 at its site of in vivo action in resolving biologically relevant 'R-loops' (**A**), displacing telomerase from 3′ ssDNA ends at telomeres or DSBs (**B**), unwinding G4 structures on the 3′ ssDNA tails at telomeres or DSBs (**C**), and unwinding G4 structures on the lagging and/or leading strands during DNA replication (**D** and **E**).

flanking dsDNA regions intact. This intrinsic property of a Pif1 monomer would suppress G4 formation until other DNA metabolic enzymes take over without causing the side effect of uncontrolled dsDNA unwinding.

Additionally, our results provide an example in which separation of helicase function can be achieved by the assembly state. Many SF1 and SF2 helicases, including PcrA and UvrD, display limited dsDNA unwinding activity as monomers, and their helicase activity can be greatly enhanced through self-assembly or interactions with accessory proteins (*Singleton et al., 2007*; *Lohman et al., 2008*). Our data suggest that the monomeric form of Pif1 is able to unwind RNA–DNA heteroduplexes and G4 structures, whereas dsDNA unwinding requires the binding of multiple Pif1 molecules per DNA substrate. As a result, DNA binding-induced enzyme oligomerization (*Barranco-Medina and Galletto, 2010*) and/or interactions with accessory proteins might play a role in regulating the translocation vs helicase activity of Pif1.

Thus far, more than 20 helicases have been shown to bind and/or unwind G4 structures in vitro, among which Pif1 displays the highest specificity in G4 unwinding (*Paeschke et al., 2013*). We have for the first time resolved the intermediates for G4 unwinding by a helicase. The three discrete steps observed are likely due to unraveling of G4 DNA one strand at a time (*Figure 7D*), which indicates that the G4 unwinding by Pif1 is tightly coupled to its translocation activity. It has been shown that the stabilization energy of a Hoogsteen GG base pair is comparable to that of AT base pair (*Mashimo and Sugiyama, 2007*), and that the lifetime of G-triplex and G-hairpin could be as long as 0.1–1 s, consistent with our observations (*Li et al., 2013*). In addition, we also showed that a G4 stabilizing compound, BRACO19, cannot inhibit Pif1-mediated G4 unwinding.

In conclusion, our study provides the mechanistic details on how Pif1 may achieve its distinct in vivo functions at the molecular level. The periodic patrolling activity by its monomeric form may keep the enzyme at its sites of in vivo action in displacing telomerase from 3′ ssDNA ends, resolving biologically relevant 'R-loops', and keeping G4 DNA sequences unfolded during DNA replication, recombination and repair. The inability of a Pif1 monomer to display dsDNA unwinding activity might be important for preventing Pif1 monomers from unwinding dsDNA after the Pif1 monomer has displaced proteins, removed RNA strands or resolved G4 structures from within a ssDNA gap.

## Materials and methods

### DNA/RNA sequences and annealing procedures

Partial duplex (pd) DNA substrates with a 3′ Poly(dT) ssDNA overhang

1. 5′-/Cy3/GCC TCG CTG CCG TCG CCA/biotin/-3′
2. 5′-TGG CGA CGG CAG CGA GGC $(T)_{15}$/iAmMC6T/$(T)_{24}$-3′
3. 5′-/Cy5/GCC TCG CTG CCG TCG CCA/biotin/-3′
4. 5′-TGG CGA CGG CAG CGA GGC $(T)_N$/Cy3/-3′, where $N$ = 21, 32, 40, 56, 72
5. 5′-GCC TCG CTG CCG TCG CCA/biotin/-3′
6. 5′-TGG CGA CGG CAG CGA GGC $(T)_{39}$/iAmMC6T/$(T)_{31}$/3AmMO/-3′
7. 5′-/Cy5/GCC TCG CTG CCG TCG CCA-3′

   Forked DNA substrates

1. 5′-$(dT)_{19}$/iAmMC6T//GCC TCG CTG CCG TCG CCA/biotin/-3′
2. 5′-TGG CGA CGG CAG CGA GGC $(T)_{40}$/Cy3/-3′

   Gapped substrates

1. 5′-/Cy5/GCC TCG CTG CCG TCG CCA/biotin/-3′
2. 5′-TGG CGA CGG CAG CGA GGC $(T)_{40}$ GCA CTC GGA TCA CCA TGG CGG ACT CTC TGC T-3′
3. 5′-AGC AGA GAG TCC GCC ATG GTG ATC CGA GTG C /Cy3/-3′
4. 5′-/Cy3/AGC AGA GAG TCC GCC ATG GTG ATC CGA GTG C-3′
5. 5′-AGC AGA GAG UCC GCC AUG GUG AUC CGA GUG C /Cy3/-3′ (RNA)
6. 5′-TGG CGA CGG CAG CGA GGC $(T)_{31}$/iAmMC6T/$(T)_{18}$ CAC CAT GGC GGA CTC TCT GCT-3′
7. 5′-AGC AGA GAG TCC GCC ATG GTG/3AmMO/-3′
8. 5′-GCC TCG CTG CCG TCG CCA/biotin/-3′
9. 5′-TGG CGA CGG CAG CGA GGC $(T)_{20}$ CAC CAT GGC GGA CTC TCT GCT-3′

Intramolecular G4 substrates (for Pif1)

1. 5′-GCC TCG CTG CCG TCG CCA/biotin/-3′
2. 5′-TGG CGA CGG CAG CGA GGC (T)$_{39}$/iAmMC6T/**GGG GGA GCT GGG GTA GAT GGG AAT GTG AGG G**/3AmMO/-3′
3. 5′-TGG CGA CGG CAG CGA GGC AAA **GGG GGA GCT GGG GTA GAT GGG AAT GTG AGG G** AAA/Cy3/-3′
4. 5′-TGG CGA CGG CAG CGA GGC (T)$_{39}$/iAmMC6T/A <u>GGG TTA GGG TTA GGG TTA GGG</u> TT/3AmMO/-3′
5. 5′-TGG CGA CGG CAG CGA GGC (T)$_{37}$/iAmMC6T/A <u>GGG TTA GGG TTA GGG TTA GGG</u> TCA CCA TGG CGG ACT CTC TGC T-3′
6. 5′-TGG CGA CGG CAG CGA GGC/iAmMC6T/A <u>GGG TTA GGG TTA GGG TTA GGG</u> TCA CCA TGG CGG ACT CTC TGC T-3′
7. 5′-AGC AGA GAG TCC GCC ATG GTG/3AmMO/-3′

pdDNA substrates with a 5′ Poly(dT) ssDNA overhang

1. 5′-/biotin/TGG CGA CGG CAG CGA GGC/Cy5/-3′
2. 5′-/Cy3/(dT)$_{40}$ GCC TCG CTG CCG TCG CCA-3′

Gapped substrates (for PcrA)

1. 5′-/biotin/TAC TCG CGC CGT CGC TCC G/Cy5/-3′
2. 5′-TGG CGA CGG CAG CGA GGC (T)$_{40}$ CGG AGC GAC GGC GCG AGT-3′
3. 5′-/Cy3/GCC TCG CTG CCG TCG CCA-3′

Intramolecular G4 substrates (for PcrA)

1. 5′-/biotin/TGG CGA CGG CAG CGA GGC/Cy5/-3′
2. 5′-/5AmMC6/TA <u>GGG TTA GGG TTA GGG TTA GGG</u>/iAmMC6T/(T)$_{39}$ GCC TCG CTG CCG TCG CCA-3′

DNA strands were purchased from Integrated DNA Technologics, Coralville, IA. The TP sequence is bold, and the TEL sequence is underlined. The amine-modified thymine (/iAmMC6T/), 3′ amino-modification (/3AmMO/), and 5′ amino-modification (/5AmMC6/) shown in the sequence enable the oligonucleotides to be labeled with the monofunctional NHS ester form of Cy3 or Cy5 dye (GE Healthcare, Piscataway, NJ). /Cy3/ and /Cy5/ represent Cy3 and Cy5 dyes, respectively, linked to the DNA backbone using phosphoramidite chemistry. The substrates were annealed by mixing ~5 μM of each strand in 10 mM Tris:HCl (pH 8.0) and 200 mM KCl, followed by slow cooling from 90°C to room temperature for ~2 hr.

## Protein expression and purification

The *S. cerevisiae* Pif1 was purified as previously reported (*Boule et al., 2005*). The *B. stearothermophilus* PcrA was purified as previously described (*Niedziela-Majka et al., 2007*; *Park et al., 2010*).

## Sample preparation and data acquisition

All smFRET experiments were performed with total internal reflection fluorescence (TIRF) microscopy (*Roy et al., 2008*) at 22 ± 1°C in imaging buffer composed of 20 mM Tris:HCl (pH 7.5), 5 mM MgCl$_2$, 60 mM KCl, 0.1 mg/ml BSA, 1 mM DTT, 2% (vol/vol) glycerol, and an oxygen scavenging system (0.5% wt/vol glucose, 3 mM Trolox, 165 U/ml glucose oxidase, and 2170 U/ml catalase) unless specified otherwise. Cy3-Cy5 labeled pdDNA substrates (50–100 pM) were immobilized on a quartz slide surface coated with polyethyleneglycol (mPEG-SC, Laysan Bio, Arab, AL) in order to eliminate non-specific surface adsorption of proteins (*Ha et al., 2002*; *Roy et al., 2008*). Surface immobilization was mediated by biotin–Neutravidin interactions between biotinylated DNA, Neutravidin (Thermo Scientific, Newington, NH), and biotinylated polymer (Bio-PEG-SC, Laysan Bio, Arab, AL). To ensure no more than one Pif1 monomer is loaded per DNA (i.e., under monomer conditions), 10 nM Pif1 was incubated with the surface-immobilized DNA for 2 min, and excess unbound proteins were then flushed out from the sample chamber using five chamber volumes of imaging buffer. Next, imaging buffer devoid of proteins was added with 20 μM ATP (unless specified otherwise) into the chamber. Finally, the Cy3/Cy5 fluorescence intensities from single DNA molecules were recorded with 30-ms

time resolution. For some experiments, ATP was added together with proteins into the chamber for data acquisition after immobilizing the DNA substrates. In the pull-down assays, Histidine$_6$-tagged Pif1 was pulled down to the surface using an antibody against the Histidine$_6$-tag following the procedure described previously (*Jain et al., 2011*). To study the unwinding of dsDNA or RNA–DNA hybrids, gapped substrates were immobilized on the PEG surface, and the mean Cy3 spot count per image (imaging area = ~2500 µm$^2$) was then determined from images taken from 5–10 different slide regions at different times after introducing ATP.

### Single-molecule data analysis

Apparent FRET efficiency was calculated from the fluorescence intensities of the donor ($I_D$) and acceptor ($I_A$) using the formula $E_{FRET} = I_A/(I_A + I_D)$. The background and the cross-talk between the donor and acceptor were considered as previously described (*Roy et al., 2008*). Analysis of individual FRET-time traces was performed using Origin or programs written in Matlab. Single molecule FRET histograms were generated by averaging for the time period of 0.15 s from ~10,000 DNA molecules. The time period Pif1 takes for one cycle of looping, $\Delta t$, was determined by visually picking the moments when $E_{FRET}$ reaches its maximum value (i.e., the peaks) in the single molecule FRET-time traces showing periodic sawtooth or wave patterns. The time period Pif1 takes for one cycle of G4 unwinding, $\Delta t$, was also determined by visually picking the moments when $E_{FRET}$ recovers to the $E_{FRET}$ value that represents folded G4 structures (*Figure 5A*). The $\Delta t$ histograms were generated by collecting $\Delta t$ from >50 molecules. To quantify the stepwise patterns observed in single molecule FRET-time traces for G4 unwinding, average $E_{FRET}$ values and the dwell time of each unwinding step were determined using the automated step-finding method used in previous single molecule FRET studies (*Kerssemakers et al., 2006*; *Myong et al., 2007*; *Lee et al., 2012*).

## Acknowledgements

We thank J Park for a generous gift of purified PcrA, and S Myong for a generous gift of BRACO19. We thank T Lohman for carefully reading the manuscript and commenting on it. These studies were supported by grants from the National Institutes of Health (GM065367 to T.H. and GM26938 to VZ), the National Science Foundation (0822613 to TH) and the American Cancer Society (PF-10-145-01-DMC to MB). TH is an employee of the Howard Hughes Medical Institute.

## Additional information

### Funding

| Funder | Grant reference number | Author |
|---|---|---|
| National Institutes of Health | GM065367 | Ruobo Zhou, Jichuan Zhang, Taekjip Ha |
| National Institutes of Health | GM26938 | Virginia A Zakian |
| National Science Foundation | 0822613 | Ruobo Zhou, Taekjip Ha |
| American Cancer Society | PF-10-145-01-DMC | Matthew L Bochman |

The funders had no role in study design, data collection and interpretation, or the decision to submit the work for publication.

### Author contributions

RZ, Conception and design, Acquisition of data, Analysis and interpretation of data, Drafting or revising the article; JZ, Conception and design, Acquisition of data, Analysis and interpretation of data; MLB, Expressed and purified Pif1 protein, Conception and design, Drafting or revising the article; VAZ, TH, Conception and design, Drafting or revising the article

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
