## [Decision Letter]

Thank you for sending your work entitled “Periodic DNA patrolling underlies diverse functions of Pif1 on R-loops and G-rich DNA” for consideration at *eLife*. Your article has been favorably evaluated by a Senior editor and 3 reviewers, one of whom is a member of our Board of Reviewing Editors.

The Reviewing editor and the other reviewers discussed their comments before reaching this decision, and the Reviewing editor has assembled the following comments to help you prepare a revised submission. Please consider each of these points, and provide either a response with additional experimental data or a detailed reasoning that addresses the issue in question.

This manuscript describes single-molecule fluorescence experiments that visualize the activity of the Pif1 DNA helicase. The authors show that it patrols ssDNA by translocating in a 5'-3'-directed fashion away from a ds-ssDNA junction in a cyclic manner. While doing so, it is able to efficiently unwind RNA–DNA hybrids and G4 quadruplex structures. The data are very convincing and their analyses rigorous. The level of kinetic detail obtained with these experiments is remarkable. The following points raised by the reviewers should be addressed in a revised manuscript and cover letter.

*Reviewer 1*
*specific comments:*

1) It would be nice to add a figure to illustrate how Pif1 might remove G4 structures in a cellular context. G4s are most likely to form on the lagging strand template, which is rendered temporarily single stranded during DNA replication. Pif1 might patrol from the 5' end of one Okazaki fragment towards a downstream G4 structure, thereby allowing completion of the next Okazaki fragment whose 3' end will have collided with the G4 from the other side.

2) Figure 3—figure supplement 1 requires a no-protein control to show that when Cy3 is placed near the ssDNA-dsDNA junction, protein-induced fluorescence enhancement can be observed. Without this, the absence of periodic increases in Cy3 fluorescence cannot be interpreted as stable binding of Pif1 at the junction.

3) The experiment in Figure 3 should be repeated with the Cy3 label placed on the 5' end of the oligo, where it is unlikely to interfere with DNA unwinding.

4) When patrolling on the DD31 and RD31 substrates, Pif1 does not seem to release abruptly (Figure 3), in contrast to the behavior on substrates presented in Figures 1 and 2. Can the authors explain this?

5) It was not entirely clear to me how the experiment in Figure 7 shows that the efficiency of unwinding is 100 %. How does this experiment show unwinding at all?

6). In the main text (Figure 6, bottom), the purpose of the experiment presented in Supplementary Figure 8 was not well explained.

*Reviewer 2*
*specific comments:*

1) Figure 1—figure supplement 1, Figure 1—figure supplement 2 and Figure 1—figure supplement 3 is confusing in terms of the presence of ATP. Figure 1 is described in the text as a broadened FRET histogram in the presence of ATP, but the 7.2 nM derived from Figure 1—figure supplement 1, Figure 1—figure supplement 2 and Figure 1—figure supplement 3 (and used in main Figure 1) is discussed in the paragraph before, suggesting that this value is obtained in the absence of ATP. The authors should be more explicit about the conditions used for the various figures.

2) The authors determine the equilibrium dissociation constant in the absence of ATP to be 7.2 nM and the dissociation rate to be less than (30 minutes)^-1^. These two values suggest an on-rate constant of 7x10^4^ m^-1^s^-1^. The experiments are performed by incubating the DNA constructs for 2 minutes with 10 nM of Pif1. However, these numbers suggest that less than 10 % of the DNA substrates will be bound. This value is inconsistent with Figures 1 and 3 (and other figures), which suggest an occupancy close to 100 %. The authors should explain this issue.

3) In Figure 2, the authors show a histogram of cycle times that they fit with a gamma distribution to obtain the number of steps in each cycle. However, with such large numbers of steps, the gamma distribution becomes very narrow (as the data indeed show) and its width starts to become dominated by the error in determining the cycle time. The authors should provide a rigorous treatment of this error and show that the width of the distribution is truly dominated by the spread in cycling times.

4) In Figure 5, the authors show the cycling time in the presence of a G4 quadruplex. The authors point out that the transition to highFRET values corresponds to the refolding of the quadruplex. Doesn't that mean that the time needed for the G4 sequence to refold is included in the measured cycling time and that the actual cycling time may be shorter?

5) The data showing the intermediate steps in unwinding a quadruplex are quite remarkable. However, this reviewer is surprised by the observation that upon unwinding the first few stretches of the quadruplex, the remaining structure is still stable enough for subsequent unwinding a second or so later. The authors should comment on this point and particularly address whether the existing information on folding energetics and kinetics would allow such a prolonged stability of the partial structures.

*Reviewer 3 specific*
*comments:*

1) There are discrepancies with reported literature that is not properly addressed in the paper. While the DNA patrolling translocation speeds on dT tails are consistent with recently reported speeds from ensemble studies (Galletto 2013; Ramanagoudr-Bhojappa 2013), the DNA/DNA unwinding rates are in disagreement with the recent report (Ramanagoudr-Bhojappa 2013). For RNA/DNA Pif1 takes about 2 min at saturating ATP to unwind 31 bp and on DNA/DNA it requires very high Pif1 and much longer times. This is puzzling because Ramanagoudr-Bhojappa 2013 showed that 16 bp DNA/DNA is unwound in 0.3 s, with an overhang of 7T that would bind one Pif1 and the rates of translocation on single stranded DNA are the same as the rates of DNA/DNA unwinding abut 70 base pairs per second. The authors need to provide an explanation for this discrepancy between their slow rates and published fast rates of unwinding.

2) Is the patrolling activity simply a consequence of having a DNA end (end of dT overhang)? Such structures would not be present in the cell raising the question of its physiological relevance.

3) There is a lack of discussion on how this activity obtained under the specific conditions used in this assay would be physiologically relevant. Pif1 initiates in all the experiments at the 3'-ss-dsDNA junction. This raises two questions, firstly, whether Pif1's translocation properties are a consequence of starting at such a junction, in other words, does Pif1 translocate in the same way in the absence of such a junction? And secondly, where would such ends be present in the genome to initiate Pif's patrolling activity?

---

## [Author Response]

Reviewer 1 specific comments:

*1) It would be nice to add a figure to illustrate how Pif1 might remove G4 structures in a cellular context. G4s are most likely to form on the lagging strand template, which is rendered temporarily single stranded during DNA replication. Pif1 might patrol from the 5' end of one Okazaki fragment towards a downstream G4 structure, thereby allowing completion of the next Okazaki fragment whose 3' end will have collided with the G4 from the other side*.

We have prepared a figure (new Figure 8) according to the reviewer’s suggestion, where we summarize several possible cellular sites for the Pif1’s patrolling activity. Pif1 may use its patrolling activity to help remove G4 structures, telomerase, and R-loops.

*2)*
Figure 3—figure supplement 1
*requires a no-protein control to show that when Cy3 is placed near the ssDNA-dsDNA junction, protein-induced fluorescence enhancement can be observed. Without this, the absence of periodic increases in Cy3 fluorescence cannot be interpreted as stable binding of Pif1 at the junction*.

We have added a no-protein control for Figure 3—figure supplement 1 (new Figure 1—figure supplement 3). We compared the average Cy3 intensity before and after adding 10 nM Pif1. Cy3 intensity increased by ∼20 % upon the addition of 10 nM Pif1, indicating Pif1 indeed binds.

*3) The experiment in*
Figure 3
*should be repeated with the Cy3 label placed on the 5' end of the oligo, where it is unlikely to interfere with DNA unwinding*.

We have repeated the experiment in Figure 3 (new Figure 3—figure supplement 1) and obtained the same result.

*4) When patrolling on the DD31 and RD31 substrates, Pif1 does not seem to release abruptly (*Figure 3*), in contrast to the behavior on substrates presented in*
Figures 1 and 2*. Can the authors*
*explain this*?

The Cy3 labeling position on DD31 and RD31 is different from that of the DNA substrates used in Figures 1 and 2. In Figures 1 and 2, Cy3 is attached to the very end of the Poly(dT) tail, whereas Cy3 is attached to a stranded complementary to the tail. When Pif1 is approaching the ss-dsDNA junction, the angle between the ssDNA region and the distal dsDNA may change (see the Figure 9 below).Author response image 1.

*5) It was not entirely clear to me how the experiment in*
Figure 7
*shows that the*
*efficiency of unwinding is 100 %. How does this experiment show unwinding at all*?

This is because the Supplementary Figure 7B (new Figure 5—figure supplement 2) shows that periodic patrolling occurs on the same DNA structure, indicating that each patrolling cycle results in G4 unwinding that we show in Figure 5. We reworded the sentence to make this point clearer.

*6). In the main text (*Figure 6*, bottom), the purpose of the experiment presented in Supplementary Figure 8 was not well explained*.

The purpose was to further confirm that the G4 structure forms under our solution condition. Although we did observe high FRET value for TP-S1 and TEL, some secondary ssDNA structures other than G4 structures can also explain the high FRET value. Because G4 formation is known to depend on the monovalent cations and the order of cations’ ability to stabilize the G4 sturcture is: K^+^>Na^+^>Li^+^, we argue that if we could observe the G4 stabilization in the same order (K^+^>Na^+^>Li^+^), the secondary structure which yielded high FRET is very likely the G4 structure. We have revised our text accordingly.

Reviewer 2 specific comments:

*1)*
Figure 1—figure supplement 1, Figure 1—figure supplement 2 and Figure 1—figure supplement 3
*is confusing in terms of the presence of ATP.*
Figure 1—figure supplement 1, Figure 1—figure supplement 2 and Figure 1—figure supplement 3
*is described in the text as a broadened FRET histogram in the presence of ATP, but the 7.2 nM derived from*
Figure 1—figure supplement 1, Figure 1—figure supplement 2 and Figure 1—figure supplement 3
*(and used in main*
Figure 1*) is discussed in the paragraph before, suggesting that this value is obtained in the absence of ATP. The authors should be more explicit about the conditions used for the various figures*.

We have added a more detailed description to Figure 1 and Figure 1—figure supplement 1 to clarify the ATP concentrations we used for each figure.

*2) The authors determine the equilibrium dissociation constant in the absence of ATP to be 7.2 nM and the dissociation rate to be less than (30 minutes)*^*-1*^*. These two values suggest an on-rate constant of 7x10*^*4*^
*m*^*-1*^*s*^*-1*^*. The experiments are performed by incubating the DNA constructs for 2 minutes with 10 nM of Pif1. However, these numbers suggest that less than 10 % of the DNA substrates will be bound. This value is inconsistent with*
Figures 1 and 3
*(and other figures), which suggest an occupancy close to 100 %. The authors should explain this issue*.

This is a good point. We think we actually overestimated the equilibrium dissociation constant (K_D_). In fact, 7 nM was derived from the single molecule (sm) FRET histograms shown in Figure 1—figure supplement 1, and the smFRET histogram at each Pif1 concentration was obtained after 2 min incubation with Pif1. In other words, the measurements were done before the system achieved the equilibrium. The true K_D_ value should be smaller than 7 nM. We have removed the mention of K_D_ in the revised manuscript. We revised the first paragraph in the Results section.

*3) In*
Figure 2*, the authors show a histogram of cycle times that they fit with a gamma distribution to obtain the number of steps in each cycle. However, with such large numbers of steps, the gamma distribution becomes very narrow (as the data indeed show) and its width starts to become dominated by the error in determining the cycle time. The authors should provide a rigorous treatment of this error and show that the width of the distribution is truly dominated by the spread in cycling times*.

The reviewer is correct in that errors in determining the Δt values would broaden the Δt histogram. However, such broadening effect would result in the underestimation of the n value ((Δ*t*)^*n*-1^exp(–*k*·Δ*t*)) because lower n values generally correspond to broader distributions in the gamma distribution. This is probably why we obtained *n*=26±3 for 32 steps on (dT)_32_ (26 is smaller than 32). An underestimation of *n* would result in overestimation of the step size. In our case, the step size estimated by diving 32 nt by 26 steps is 1.2 nt per step, and considering the likelihood of overestimation due to errors, the true step size must be 1 nt. If the kinetic step were 2 nt, for example, the broadening effect would have resulted in a n value of less than 16 for (dT)_32_ which was not the case. We have revised the text to clarify this point.

*4) In*
Figure 5*, the authors show the cycling time in the presence of a G4 quadruplex. The authors point out that the transition to highFRET values corresponds to the refolding of the quadruplex. Doesn't that mean that the time needed for the*
*G4 sequence to refold is included in the measured cycling time and that the actual cycling time may be shorter*?

Yes, we agree with the reviewer. The time needed for the G4 sequence to refold is indeed included in the measured cycling time (Δ*t*). In Figure 7, we further divided Δ*t* into five portions: Δ*t*_*1*_*,* Δ*t*_*2*_*,* Δ*t*_*3*_*,* Δ*t*_*4*_*,* and Δ*t*_*5*_*,* in which Δ*t*_*5*_ represents the G4 refolding time.

*5) The data showing the intermediate steps in unwinding a quadruplex are quite remarkable. However, this reviewer is surprised by the observation that upon unwinding the first few stretches of the quadruplex, the remaining structure is still stable enough for subsequent unwinding a second or so later. The authors should comment on this point and particularly address whether the existing information on folding energetics and kinetics would allow such a prolonged stability of the partial structures*.

1) Previous MD simulation results have suggested G-hairpin and G-triplex are two relatively stable intermediates along the G4 folding pathway ([26] J Am Chem Soc 132(42):14910-14918). It was found that the stabilization energy of a Hoogsteen GG base pair was less than that of a GC base pair but is comparable to that of AT base pair (Mashimo et al. 2007 Nucleic Acids Symp Ser (Oxf) (51) 239-240. 2) The lifetime of G-triplex and G-hairpin could be as long as 0.1–1 seconds, according to previous experiments. For example, previous optical tweezers experiments have identified G-triplex as a stable intermediate [21] J Am Chem Soc 135(17):6423-6426: when 32 pN was applied to the two ends of a G4 sequence, the lifetimes of the fully folded state and the G-triplex state were 0.54 s and 0.52 s, respectively. Another example is the unwinding of a small DNA hairpin by a protein migrating along the DNA strand ([36] Nature 461(7267):1092-1097). The small hairpin only consists of 7 bp stem and 3 nt loop and an intermediate state with a lifetime of ∼0.90 sec was observed along the unwinding pathway. Given that GG base pairs have comparable stabilization energy, one would expect a comparably long lifetime for G-hairpins. We have revised the text to clarify this point.

Reviewer 3 specific comments:

*1) There are discrepancies with reported literature that is not properly addressed in the paper. While the DNA patrolling translocation speeds on dT tails are consistent with recently reported speeds from ensemble studies (Galletto 2013; Ramanagoudr-Bhojappa 2013), the DNA/DNA unwinding rates are in disagreement with the recent report (Ramanagoudr-Bhojappa 2013). For RNA/DNA Pif1 takes about 2 min at saturating ATP to unwind 31 bp and on DNA/DNA it requires very high Pif1 and much longer times. This is puzzling because Ramanagoudr-Bhojappa 2013 showed that 16 bp DNA/DNA is unwound in 0.3 s, with an overhang of 7T that would bind one Pif1 and the rates of translocation on single stranded DNA are the same as the rates of DNA/DNA unwinding abut 70 base pairs per second. The authors need to provide an explanation for this discrepancy between their slow rates and published fast rates of unwinding*.

The discrepancy is most likely due to the difference in assembly state and the way the experiments were performed. In Ramanagoudr-Bhojappa et al., 200 nM Pif1 was pre-incubated with 2 nM DNA before initiating the unwinding reaction by introducing ATP. In contrast, we added 10 or 100 nM Pif1 together with ATP to the DNA-immobilized sample chamber. There are two possible reasons that may lead to the apparent discrepancy: 1) It was not clear how many Pif1 monomers were loaded per DNA molecule in Ramanagoudr-Bhojappa et al., and in fact it is quite likely that more than one Pif1 monomers were loaded per DNA molecule which would lead to efficient unwinding of DNA-DNA duplexes. Under our conditions, we know from the single molecule time traces that at 10 nM Pif1, only one Pif1 monomer is loaded (no unwinding observed) and 100 nM Pif1 results in multiple Pif1 loading per DNA (unwinding observed). 2) In Figure 3, the rate-limiting step there for unwinding reaction was the second Pif1 monomer binding to DNA. As a result, 100 nM Pif1 took several minutes to unwind DNA-DNA duplexes at 1 mM ATP. However, in Ramanagoudr-Bhojappa et al., which pre-assembled the helicase on the DNA the 0.3 sec did not include the binding time of Pif1 at all. Another possibility is a limited processivity of DNA unwinding by Pif1 oligomer. It may unwind 16 bp rapidly whereas 31 bp may require multiple attempts, hence slower. We added the following sentence to the manuscript: “The dsDNA unwinding speed previously reported was much higher (∼ 0.3 second for unwinding 16 bp dsDNA) (34) either because the helicase was preassembled on the DNA before introducing ATP in that study, or due to a limited processivity of dsDNA unwinding by Pif1 oligomer.”

*2) Is the patrolling activity simply a consequence of having a DNA end (end of dT overhang)? Such structures would not be present in the cell raising the question of its physiological relevance*.

We don’t think so. Pif1’s patrolling activity was observed not only on partial DNA duplexes containing a 3’ DNA overhang but also on gapped DNA substrates (Figure 3) and Y-shaped DNA substrates (Figure 1) that do not have a free ssDNA end. In addition, the partial DNA duplexes containing a 3’ DNA overhang should indeed be present in cells. For example, telomeres and double stranded DNA breaks both contain such a 3’ ssDNA end (see new Figure 8 that we added in response to Reviewer 1). Furthermore, we showed that Pif1 uses its patrolling activity to remove RNA/DNA hybrids (i.e., R-loops) and G4 structures, two physiological relevant structures present in vivo. The presence of R-loops and G4 structures, if not resolved in a timely manner, would result in genome instability including replication pauses, mutations, recombination, chromosomal rearrangements and chromosome loss ([1] Mol Cell 46(2):115-124, [31] Cell 145(5):678-691, [30] Nature 497:458-462).

*3) There is a lack of discussion on how this activity obtained under the specific conditions used in this assay would be physiologically relevant. Pif1 initiates in all the experiments at the 3'-ss-dsDNA junction. This raises two questions, firstly, whether Pif1's translocation properties are a consequence of starting at such a junction, in other words, does Pif1 translocate in the same way in the absence of such a junction? And secondly, where would such ends be present*
*in the genome to initiate Pif's patrolling activity*?

The presence of a 3'-ss-dsDNA junction is not a prerequisite for Pif1 to start translocation along ssDNA, because Pif1 translocation was detected when such a junction is absent ([12] Nucleic Acids Res) ^8^. 3'-ss-dsDNA junctions are present in many DNA intermediates during replication, transcription and recombination, as well as in telomeres (see new Figure 8). Therefore, the patrolling activity we report in this manuscript could take part in many scenarios where the 3’-ss-dsDNA junctions are present.